# Synthetic Cyclic C_5_-Curcuminoids Increase Antioxidant Defense and Reduce Inflammation in 6-OHDA-Induced Retinoic Acid-Differentiated SH-SY5Y Cells

**DOI:** 10.3390/antiox14091057

**Published:** 2025-08-28

**Authors:** Edina Pandur, Gergely Gulyás-Fekete, Győző Kulcsár, Imre Huber

**Affiliations:** 1Department of Pharmaceutical Biology, Faculty of Pharmacy, University of Pécs, Rókus str. 4, H-7624 Pécs, Hungary; 2Department of Biochemistry and Medical Chemistry, Medical School, University of Pécs, Szigeti str. 12, H-7624 Pécs, Hungary; gergely.gulyas@aok.pte.hu; 3Institute of Pharmaceutical Chemistry, Faculty of Pharmacy, University of Pécs, Rókus str. 4, H-7624 Pécs, Hungary; gyozo.kulcsar@aok.pte.hu (G.K.); imre.huber@aok.pte.hu (I.H.)

**Keywords:** Parkinson’s disease, oxidative stress, synthetic cyclic C_5_-curcuminoids, cyclic chalcones, antioxidant, apoptosis, inflammation

## Abstract

Parkinson’s disease (PD) is recognized as one of the most common neurodegenerative disorders globally. The primary factor contributing to this condition is the loss of dopaminergic neurons, which results in both motor and nonmotor symptoms. The etiology of neurodegeneration remains unclear. However, it is characterized by the elevated production of reactive oxygen species, which subsequently leads to oxidative stress, lipid peroxidation, mitochondrial dysfunction, and inflammation. The investigation of the applicability of natural compounds and their derivatives to various diseases is becoming increasingly important. The possible role of curcumin from *Curcuma longa* L. and its derivatives in the treatment of PD has been partially investigated, but there are no data on the action of synthetic cyclic C_5_-curcuminoids and chalcones tested in a Parkinson’s model. Two chalcones and five synthetic cyclic C_5_-curcuminoids with potential antioxidant properties were investigated in an in vitro model of 6-hydroxydopamine (6-OHDA)-induced neurodegeneration in differentiated SH-SY5Y cells. Reactive oxygen species (ROS) production, total antioxidant capacity, antioxidant enzyme activity, thiol and ATP levels, caspase-3 activity, and cytokine release were examined after treatment with the test compounds. Based on these results, one cyclic chalcone (compound **5**) and three synthetic cyclic C_5_-curcuminoids (compounds **9**, **12**, and **13**) decreased oxidative stress and apoptosis in our in vitro model of neurodegeneration. Compounds **5** and **9** were also successful in decreasing the production of pro-inflammatory cytokines (IL-6, IL-8, and TNF-α), while promoting the release of anti-inflammatory cytokines (IL-4 and IL-10). These findings indicate that these two compounds exhibit potential antioxidant, anti-apoptotic, and anti-inflammatory properties, rendering them promising candidates for drug development.

## 1. Introduction

Neurodegenerative disorders are emerging diseases worldwide. According to the latest estimates, in two decades, the number of people with Parkinson’s disease (PD) will reach 14.2 million [1]. PD is characterized by motor symptoms such as muscle rigidity, tremors at rest, and balance issues, as well as nonmotor symptoms like cognitive challenges, autonomic system issues, and depressive disorders [2].

The etiology of PD remains incompletely elucidated. Factors such as aging, genetic predispositions, environmental influences, and social determinants, including lifestyle and stress, contribute to the pathogenesis of the disease [3]. Approximately 10–15% of individuals with PD, particularly those with early-onset forms, exhibit genetic variations in PD-associated genes, notably *GBA1* and *LRRK2*, which are significant genetic risk factors [4]. The pathology of PD is characterized by the degeneration of dopaminergic neurons, leading to a reduction in dopamine synthesis. Oxidative stress, which contributes to mitochondrial dysfunction leading to energy stress and iron accumulation, initiates ROS production and lipid peroxidation, resulting in ferroptosis [5,6,7,8,9].

Treatments of PD aim to restore dopamine levels using dopamine receptor agonists, inhibitors of dopamine-degrading enzymes, monoamine oxidase (MAO)-B inhibitors, and levodopa, which is metabolized to dopamine [10,11,12,13]. Another possibility is invasive deep brain stimulation and Duopa™ therapy to decrease tremors [14,15].

In the field of pharmaceutical research, natural compounds derived from herbs often serve as essential sources of lead molecules and potential drug candidates [16,17]. Species in the *Zingiberaceae* (ginger) family, such as *Curcuma longa* L. and *Curcuma domestica* L., are the parent plants of turmeric (Curcuma), containing curcumin (I) as the major component (Figure 1). It exhibits a wide range of advantageous biological properties, such as antimicrobial, antifungal, anti-inflammatory, antioxidant, neuroprotective, cytotoxic, and antitumor effects [18]. The antiradical and antioxidant activities of curcumin (I) have been extensively investigated [19]. The antioxidant ability of curcumin is due to both the phenolic aromatic rings and the 1,3-dicarbonyl moiety. Monocarbonyl C_5_-curcumin (II) is a component of turmeric (Figure 1) and has strong antioxidant properties [19]. Truncated C_5_-curcumin (II) can be synthesized from curcumin (I) via pyrolysis [20]. Another chemically closely related group of natural compounds with one enone motif in their structure is the large group of chalcones (III, Figure 1). Chalcones are found in many plants and, like curcumin, have demonstrated a number of beneficial biological effects. Synthetic cyclic chalcones also possess advantageous pharmacological properties [21].

Owing to the chemical and biochemical shortcomings of curcumin (I), there is an ongoing interest in the synthesis of more stable and effective cyclic analogs of C_5_-curcuminoids (V) [22,23]. Their pleiotropic feature may be responsible for the coexisting antioxidant, multiple drug resistance (MDR) revertant, and other beneficial properties of these derivatives [24]. However, studies on the antioxidant abilities of cyclic C_5_-curcuminoids are limited.

The molecules, including A and its derivatives, feature a unique –NOH function in their structure (Figure 2), which offers antioxidant protection to healthy tissues while still enabling the compounds to retain their toxicity against cancer cells. This special feature provides selective toxicity to heterocyclic C_5_-curcuminoid compounds [25]. Another study reported that C_5_-curcuminoids, such as cyclic C_5_-curcuminoid B conjugated with a polyamine function on the central nitrogen atom (Figure 2), allowing the compounds to cross mitochondrial membranes and develop mitochondria-directed neuroprotective effects [26].

Quinoline-based antifungal cyclic C_5_-curcuminoid C (Figure 2) and its derivatives prepared in a Claisen-Schmidt condensation reaction showed better radical scavenging activity than butylated hydroxytoluene [27]. Antibacterial (*E. coli* and *S. aureus*) derivative D (Figure 2) prepared from piperidin-4-one hydrochloride and thiophene-2-carboxaldehyde proved to be a strong antioxidant because of its high free radical scavenging ability in nine different assays [28].

A review has examined the potential application of curcumin and its derivatives in addressing neurodegenerative disorders, including Alzheimer’s and Parkinson’s diseases, as well as brain malignancies [29]. Another article reviewed three clinical trials, indicating that curcumin may provide modest benefits as an adjunctive treatment for patients with PD. These findings are preliminary and necessitate further trials for validation [30]. We did not identify any publications concerning synthetic cyclic C_5_-curcuminoids tested in an antiparkinsonian context.

In this study, we investigated two cyclic chalcones (**4** and **5**) and five synthetic cyclic C_5_-curcuminoids (**8**, **9**, **11**, **12**, and **13**) for their potential antioxidant properties using an in vitro model of neuronal damage induced by 6-OHDA. SH-SY5Y cells were differentiated with all-trans-retinoic acid and subsequently treated with 6-OHDA to simulate oxidative stress. These cells were then exposed to our sample compounds (**4**, **5**, **8**, **9**, **11**, **12**, and **13**), and their effects on iron accumulation, oxidative stress defense, thiol, malondialdehyde (MDA) and ATP concentrations, apoptosis, and cytokine release were assessed. The findings indicated that cyclic chalcone (**5**) and three synthetic cyclic C_5_-curcuminoids (**9**, **12**, and **13**) were effective in mitigating oxidative stress and apoptosis. Notably, only compounds **5** and **9** significantly reduced pro-inflammatory cytokine levels. Additionally, these two compounds exhibited the highest IC50 values in the in vitro model, indicating their potential as promising drug candidates.

## 2. Materials and Methods

### 2.1. Chemical Experimental Protocol

A Barnstead-Electrothermal 9100 apparatus (Thermo Fisher Scientific Inc., Waltham, MA, USA) was used to ascertain the uncorrected melting points. Silica gel 60 (0.2–0.5 mm, Merck Life Sciences Kft., Budapest, Hungary) was used for column chromatography and pre-coated silica gel 60 (F-254, Merck Life Sciences Kft., Budapest, Hungary) plates for TLC.

The nuclear magnetic resonance (NMR) 1H- and 13C-spectra were recorded using a Bruker Avance III 500 (500.15/125.77 MHz for 1H/13C) spectrometer (Bruker Corp., Ettlingen, Germany) according to our previous publication [31]. The chemical shifts were calibrated against the residual solvent signals. The measurements were conducted at a probe temperature of 298 K using a dimethyl sulfoxide (DMSO)-d6 solution. All the 1H and 13C NMR spectra were in good agreement with the expected structures.

A Thermo Dionex Ultimate 3000 liquid chromatograph (Dionex, Sunnyvale, CS, USA) coupled with a Thermo Q Exactive Focus Hybrid Quadrupole-Orbitrap mass spectrometer (Thermo Fisher Scientific, Waltham, MA, USA) was utilized for High-Performance Liquid Chromatography-Mass Spectrometry (HPLC-MS) analyses according to our previous publication [31]. For data analysis and evaluation, Q Exactive Focus 2.1 Software and Xcalibur 4.2. Software (Thermo Fisher Scientific Inc., Waltham, MA, USA) were applied. Separation of the compounds was performed using a Thermo Hypersil GOLD C18 analytical column. The mobile phase consisted of a 0.1% formic acid solution (A) and methanol containing 0.1% formic acid (B). Gradient elution was performed as follows: 0 min, 20% B; 6 min, 80% B; 8 min, 80% B; 10 min, 20% B; and 10 min stop. Chromatographic analysis was conducted at 40 °C at a flow rate of 0.3 mL/min. The injection volume was 5 μL [31].

The physical and analytical data of the synthesized compounds 4 [32], 8 and 9 [33], 11 [34], 12 [35] were identical to the original ones. The numbering of the compounds follows that of the schemes outlined in the Results section.

### 2.2. Cell Culture and Treatment

SH-SY5Y human neuroblastoma cells (ATCC-CRL-2266; LGC Ltd., Teddington, UK) were grown in DMEM/F12 medium (Biowest Ltd., Nuaillé, France) that was enriched with 10% fetal bovine serum (FBS; Biowest Ltd., Nuaillé, France), along with 1% non-essential amino acids (NEAA; Biowest Ltd., Nuaillé, France) and 1% penicillin/streptomycin (P/S; Biowest Ltd., Nuaillé, France). To differentiate SH-SY5Y cells into dopaminergic neurons, 1 µM all-trans retinoic acid (ATRA; Merck Life Sciences Kft., Budapest, Hungary) was added for 5 days in a complete culture medium with reduced serum (1% FBS) [36]. Due to its low iron content, it becomes depleted over the 5-day incubation period. The medium is replaced only after differentiation, with the serum content remaining reduced. To enhance iron transport into cells, the fresh culture medium was supplemented with 50 µM ferric ammonium citrate (FAC; Merck Life Sciences Kft., Budapest, Hungary) during 6-OHDA treatment. This supplement was also administered to the control cells to ensure comparability. Oxidative stress was induced after differentiation by adding 150 µM 6-OHDA (Merck Life Sciences Kft., Budapest, Hungary) for 24 h. The differentiated SH-SY5Y cells were then treated with seven different synthetic cyclic sample compounds for 24 h. Rasagiline (Merck Life Sciences Kft., Budapest, Hungary), a monoamine oxidase B inhibitor, was used as the positive control at a concentration of 1 µM for 24 h [37].

### 2.3. Viability Measurements

The viability of differentiated SH-SY5Y cells was assessed using a TOX8 kit (Merck Life Sciences Kft., Budapest, Hungary) following the manufacturer’s guidelines. Briefly, SH-SY5Y cells were cultured in 96-well plates at a density of 10^4^ cells. The cells were then differentiated for 5 days with ATRA. Then, the cells were treated with the sample compounds at increasing concentrations of 50 nM–1 µM and 1 µM–20 µM. The compounds were first dissolved in 100% DMSO and subsequently diluted with the suitable culture medium before being applied to the cells. Following a 24 h treatment period, 10 µL of TOX8 reagent was introduced to each well, and the mixture was incubated for an additional 2 h. The optical density was assessed at a wavelength of 600 nm using a MultiSkan GO spectrophotometer (Thermo Fisher Scientific Inc., Waltham, MA, USA) [38]. DMSO-treated cells were used as controls. The optimal 6-OHDA and rasagiline concentrations were determined as previously described [38]. The IC50 values were derived from cell viability data using GraphPad Prism 8 software (GraphPad Software, San Diego, CA, USA).

### 2.4. Determination of the Reactive Oxygen Species (ROS)

SH-SY5Y cells after differentiation were seeded in a 96-well plate at a concentration of 10^4^ cells/well and exposed to 6-OHDA treatment for 24 h. The cells were then treated with the test compounds according to their IC50 values. Three initial concentrations of each compound were used for ROS measurements (Table 1). The amount of ROS was determined in SH-SY5Y cells using a Fluorometric Intracellular ROS Kit (Merck Life Sciences Kft., Budapest, Hungary). The assay was conducted following the guidelines provided by the manufacturer. Cells were exposed to the detection reagent for 30 min in a humidified environment with 5% CO_2_ at 37 °C. The cells were incubated for 30 min with the detection reagent in a humidified atmosphere containing 5% CO_2_ at 37 °C. Fluorescence intensity was measured at an excitation wavelength of 640 nm and emission wavelength of 675 nm using an EnSight Multimode microplate reader (PerkinElmer, Rodgau, Germany). The ROS level was compared to that of DMSO-treated control cells and expressed as a percentage of the control, which was considered 100% [39].

### 2.5. Total Antioxidant Capacity (TAC) Measurement

The Antioxidant Assay Kit (Merck Life Science Kft., Budapest, Hungary) was utilized to measure the total antioxidant capacity encompassing both small-molecule and enzymatic antioxidant defenses. Differentiated SH-SY5Y cells were cultured and treated in 6-well plates at 5 × 10^5^ cells/well (Sarstedt Inc., Nümbrecht, Germany). The cells were then treated with 6-OHDA for 24 h. The cells were then treated with the test compounds according to their IC50 values, similar to the ROS measurements listed in Table 1. DMSO-treated cells were used as controls. Following incubation, SH-SY5Y cells were harvested by spinning them at 1200 rpm for 15 min. The resulting cell pellets were rinsed once with ice-cold 1× phosphate-buffered saline (PBS, Capricorn Scientific GmbH, Ebsdorfergrund, Germany). The pellets were then lysed in ice-cold PBS. After another round of centrifugation, 100 µL from each sample was taken for analysis. Optical density was measured at 570 nm using a MultiSkan GO spectrophotometer. Antioxidant concentrations were expressed as Trolox equivalents (µM) [40].

### 2.6. Glutathione Peroxidase (GPx) Activity Measurement

GPx activity was determined using a Glutathione Peroxidase Assay Kit (Merck Life Science Kft., Budapest, Hungary) according to the manufacturer’s protocol. Differentiated SH-SY5Y cells were cultured and treated in 6-well plates at 5 × 10^5^ cells/well. Cells were treated with 6-OHDA, sample compounds, and rasagiline. After incubation, SH-SY5Y cells were collected by centrifugation at 1200 rpm for 15 min. SH-SY5Y cell pellets were washed once with ice-cold 1× PBS and lysed in 200 µL of PBS by sonication. After centrifugation for 10 min at 14,000× *g*, glutathione peroxidase activity was measured using 10 µL of the supernatant from each sample. The optical density at 340 nm was determined using a Multiskan GO spectrophotometer. The activity of GPx was reported in units per liter (U/L).

### 2.7. Superoxide Dismutase (SOD) Activity Determination

SOD activity was determined using a Superoxide Dismutase Assay Kit (Merck Life Science Kft., Budapest, Hungary) according to the manufacturer’s protocol. Differentiated SH-SY5Y cells were cultured and treated in 6-well plates at 5 × 10^5^ cells/well. The cells were treated with 6-OHDA and then with the sample compounds and rasagiline. After incubation, SH-SY5Y cells were collected by centrifugation at 1200 rpm for 15 min. The cell pellets were washed once with ice-cold 1× PBS. The cells were disrupted in 500 µL of cold 1× Lysis Buffer on ice for 10 min and then collected through centrifugation. To assess SOD activity, 20 µL of the supernatant was taken from each sample. The optical density was determined at 440 nm using a Multiskan GO spectrophotometer. SOD activity was calculated following the manufacturer’s guidelines and reported in U/mL [39].

### 2.8. Total Intracellular Iron Content Determination

The intracellular iron content was determined using a colorimetric ferrozine-based assay described by Riemer et al. [41]. Differentiated SH-SY5Y cells were cultured at a density of 5 × 105 cells/well in 6-well plates. The cells were then treated with 6-OHDA for 24 h. After incubation, cells were treated with the sample compounds and rasagiline. After incubation, SH-SY5Y cells were collected by centrifugation at 1200 rpm for 15 min. The SH-SY5Y cell pellets were washed once with ice-cold 1× PBS. The cell pellets were lysed with 200 µL of 50 mM NaOH at room temperature for 2 h. Following incubation, the lysates were neutralized by adding an equal amount of 1N HCl. Subsequently, 100 µL of the sample was combined with 100 µL of an iron-releasing reagent (1.4 M HCl, 4.5% *w*/*v*) and incubated for 2 h at 60 °C. Once the incubation was complete, the samples were allowed to cool to room temperature. Then, 30 µL of an iron detection reagent was introduced, and the mixture was left to incubate at room temperature for 30 min. The optical density was measured in 96-well plates at 550 nm using a Multiskan GO spectrophotometer. A standard curve of FeCl_3_ was used to determine its concentration, which was treated in the same manner as for the samples. The total iron content measurement encompasses both free and protein-bound iron within the cells, assessed after reduction in the form of the Fe^2+^-ferrozine complex and normalized to the protein content of the cells. The protein concentration for each sample was measured using a DC Protein Assay Kit (Bio-Rad Inc., Hercules, CA, USA). The amount of iron within the cells was quantified as µM of iron/mg of protein [36].

### 2.9. Thiol Concentration Measurement

The total thiol concentration of the cells was determined using a Fluorometric Thiol Assay Kit (Merck Life Science Kft., Budapest, Hungary). Differentiated SH-SY5Y cells were cultured in 25 cm^2^ flasks at 10^6^ cells/flask (Sarstedt Inc., Nümbrecht, Germany). The cells were treated with 6-OHDA for 24 h, and then with the test compounds for 24 h. After incubation, the cells were collected by centrifugation at 1200 rpm for 15 min. The cell pellets were washed once with 1 mL ice-cold PBS and lysed in 100 µL assay buffer. The measurements were performed in 96-well plates following the manufacturer’s guidelines. For the analysis, 50 µL of both standards and samples were combined with 50 µL of the Detection Reagent in the wells. The plates were then incubated at RT for 1 h. Fluorescence intensity was recorded at excitation and emission wavelengths of 490/535 nm using the EnSight Multimode Plate Reader. The results were calculated based on the standard curve values of reduced glutathione (GSH) and expressed in µM [39].

### 2.10. Malondialdehyde (MDA) Detection

The levels of MDA were measured using the MDA Colorimetric Assay Kit (Thermo Fisher Scientific Inc., Waltham, MA, USA). Differentiated SH-SY5Y cells were cultured and treated in 25 cm^2^ flasks at 10^6^ cells/flask. After incubation, SH-SY5Y cells were collected by centrifugation. The cells were homogenized in 400 μL of PBS, and 200 μL supernatant was used for the MDA determination. The optical density was determined at 532 nm using a Multiskan GO spectrophotometer. The MDA concentration was calculated using a standard curve (0–40 μmol). The results were normalized to the protein content of the supernatant and expressed as μmol/mg protein.

### 2.11. ATP Concentration Determination

The intracellular ATP concentration of the treated cells was measured using an ATP Determination Kit (Thermo Fisher Scientific Inc., Waltham, MA, USA). Differentiated SH-SY5Y cells were cultured and treated in 6-well plates at 5 × 10^5^ cells/well. After incubation, SH-SY5Y cells were collected by centrifugation at 1200 rpm for 15 min. SH-SY5Y cell pellets were lysed in 100 µL of 1× Reaction Buffer, and 10 µL of the lysate was added to 1× Reaction Buffer in a total volume of 100 µL. The luminescence was measured using an EnSight Multimode microplate reader. Values are expressed as nM [37].

### 2.12. Cytochrome c Measurement

Cytochrome c levels were determined using a Human Cytochrome c ELISA Kit (Thermo Fisher Scientific Inc., Waltham, MA, USA), according to the manufacturer’s protocol. Differentiated SH-SY5Y cells were cultured at a density of 5 × 10^5^ cells/well in 6-well plates. The cells were exposed to 6-OHDA for 24 h. Following the incubation period, cells were treated with the sample compounds and rasagiline. After incubation, SH-SY5Y cells were harvested, and the pellets were washed once with ice-cold 1× PBS. The cells were disrupted in lysis buffer for 1 h at RT with gentle shaking and then centrifuged at 200× *g* for 15 min. The supernatants were diluted following the manufacturer’s guidelines, and 100 µL of each was utilized for ELISA. The absorbance was recorded at 450 nm using a Multiskan GO spectrophotometer. Concentrations were calculated using SkanIt software 5.1 (Thermo Fisher Scientific, Waltham, MA, USA). The results were expressed in ng/mL [42].

### 2.13. Caspase-3 Activity Assay

The activity of caspase-3 was measured using a Caspase-3 Assay Kit (Merck Life Science Kft., Budapest, Hungary), following the instructions provided by the manufacturer. Differentiated SH-SY5Y cells were cultured and treated with 6-OHDA for 24 h, then with the sample compounds or rasagiline for 24 h, in 25 cm^2^ flasks using 10^6^ cells/flask. DMSO-treated cells were used as controls. After incubation, SH-SY5Y cells were harvested, and then the cell pellets were washed once with ice-cold 1× PBS. The cells were lysed in 300 µL of ice-cold lysis buffer (50 mM HEPES, pH 7.2, 100 mM NaCl, 0.5% (*v*/*v*) Triton X-100) with shaking for 30 min on ice. Cell lysates were centrifuged at 2500× *g* for 10 min at 4 °C. Caspase-3 activity was measured in 50 µL of supernatant. The fluorescent intensity (FI) was determined at λEx 400 nm/λEm 490 nm using an EnSight Multimode Microplate Reader. The FI was normalized to the control values and expressed as ∆FI.

### 2.14. Enzyme-Linked Immunosorbent Assay (ELISA)

Following the administration of the sample compounds and rasagiline to SH-SY5Y cells, the culture media from both the control and treated groups were collected and kept at −80 °C until the analysis. Cytokine and chemokine secretion of the differentiated SH-SY5Y cells was determined using ELISA kits from Thermo Fisher Scientific Inc.: Human IL-4 ELISA Kit, Human IL-6 ELISA Kit, Human IL-8 ELISA Kit, Human IL-10 ELISA Kit, Human TNF-α ELISA Kit, and Human Fractalkine ELISA Kit. All the ELISA kits were used according to the manufacturer’s instructions. The levels of secreted proteins were expressed in pg/mL [40].

### 2.15. Statistical Analysis

The experiments were repeated three times. Five technical replicates were used for viability, TAC, and ROS measurements. Three technical replicates were performed for GPx, SOD, iron, thiol, MDA, ATP, cytochrome c, caspase-3 activity, and cytokine ELISA. Data analysis was conducted using SPSS software (version 24.0; IBM Corporation, Armonk, NY, USA). The determination of statistical significance was carried out through one-way ANOVA followed by Tukey’s post hoc test. Results are expressed as mean ± standard deviation (SD), and a *p*-value of less than 0.05 was considered to indicate statistical significance.

## 3. Results

### 3.1. Chemical Synthesis

One of our major activities in our laboratory is the synthesis and pharmacological testing of new derivatives of the cyclic or heterocyclic C5-curcuminoid family, as well as analogs, such as cyclic chalcones [31,33,43]. As a continuation of this work, we decided to prepare couples of derivatives **4** and **5**, **8** and **9** or **11** and **12** with phenolic and similar but nonphenolic substituents to test their differences in terms of antioxidant ability. Phenolic members **5** and **9** of the couples were formed in the acid-catalyzed Claisen-Schmidt condensation, while the others were obtained in a basic reaction milieu (Figure 1, Figure 2 and Figure 3). Methanol saturated with HCl(g) was used as an acid.

4-Hydroxy-cyclohexanone (**6**) is a new cyclic ketone used in the design and synthesis of C_5_-curcuminoids. We introduced it [33] to build an additional element into molecules with hydrogen donor capabilities.

In addition to homocyclic chalcone couples **4** and **5** or homocyclic C_5_-curcuminoids **8** and **9**, heterocyclic pairs **11** and **12** were also prepared to determine whether there was a difference in antioxidant activity. Derivative **13** contains an amide substituent on the central nitrogen, which was introduced by an acylation reaction using succinic anhydride. This substituent was a very good performer in our previous investigations [31] on astrocytoma and neuroblastoma cells. In addition, the structure of compound B (Figure 1) has mitochondria-directed neuroprotective characteristics.

2E-(3′,4′-Dimethoxybenzylidene)-1-tetralone (**4**) was prepared according to a previously described method for the preparation of 2-(3′,4′-methylenedioxybenzylidene)-1-tetralone in [32], starting from 1,2-dimethoxybenzaldehyde (**2**) and α-tetralone (**1**).

2-(3′,4′-Dihydroxybenzylidene)-1-tetralone (**5**): 5 mmol of α-tetralone (**1**) and 5 mmol 1,2-dihydroxybenzaldehyde (**3**) were dissolved in 30 mL of dry methanol saturated with dry hydrogen chloride gas. This solution was left to stand at ambient temperature overnight, diluted with 100 mL of water, and the separated crystals were collected on a glass filter. The crude product was recrystallized from methanol to produce fine pale-yellow crystals. Yield: 76%. Mp: 184–186 °C. 1H NMR (400 MHz, DMSO-d6) δ (ppm) 2.93 (t, J = 6.7 Hz, 2H), 3.09 (t, J = 6.2 Hz, 2H), 6.80–6.85 (m, 1H), 6.90 (dd, J = 1.8 Hz, 8.3 Hz, 1H), 6.99 (d, J = 1.8 Hz, 1H), 7.33–7.42 (m, 2H), 7.55 (dd, J = 1.2 Hz, 7.4 Hz, 1H), 7.58 (s, 1H), 7.93 (d, J = 7.7 Hz, 1H). 13C NMR (100 MHz, DMSO-d6) δ (ppm) 26.62, 27.74, 115.59, 117.19, 122.87, 126.44, 126.81, 127.16, 128.29, 132.03, 133.05, 133.11, 136.59, 143.03, 145.00, 146.72, 186.48. MS: m/z for C_17_H_14_O_3_ [M + H]+ calculated: 267.291; observed: 267.155. NMR and MS spectra are shown in Appendix A, respectively.

(2E,6E)-2,6-Bis(3′,4′-dimethoxybenzylidene)-4-hydroxycyclohexanone (**8**) was prepared according to a previously described method [33].

(2E,6E)-2,6-Bis(4′-hydroxy-3′-methoxybenzylidene)-4-hydroxycyclohexanone (**9**)was prepared according to a method described before [33].

(2E,6E)-2,6-Bis(3′,4′-dimethoxybenzylidene)-piperid-4-one (**11**) was prepared according to a previously described method [34].

(2E,6E)-2,6-Bis(4′-hydroxy-3′-methoxybenzylidene)-piperid-4-one (**12**) was prepared according to a method described before [35].

4-[(3′E,5′E)-3′,5′-bis(4′-hydroxy-3′-methoxybenzylidene)-piperid-4′-one-1′-yl]-4-oxobutane-1-carboxylic acid (**13**): was prepared from compound 12 and succinic anhydride according to a method described for other derivatives in [23]. Yield: 71%. Mp.: 205–206 °C, recrystallized from methanol. 1H-NMR (500 MHz, DMSO-D6) δ (ppm) 2.33 (m, 2H), 2.45 (m, 2H), 3.83 (m, 6H), 4.83 (m, 4H), 6.89 (m, 2H), 7.04 (m, 2H), 7.13 (s, 2H), 7.63 (m, 2H). 13C-NMR (125 MHz, DMSO-D6) δ (ppm) 27.0, 28.6, 42.5 *, 45.9 *, 55.6, 115.0, 115.6, 124.3, 125.6 *, 125.8 *, 129.4 *, 129.8 *, 136.3, 147.5, 148.4, 169.8, 173.4, 185.4 (*: doubling of signals). MS: *m*/*z* for C_25_H_25_NO_8_ [M + H]+ calculated: 468.468; observed: 468.166. NMR and MS spectra are shown in Appendix A, respectively.

### 3.2. IC50 Values

6-OHDA-mediated oxidative stress was observed in differentiated SH-SY5Y cells. The cells were treated with seven different cyclic C_5_-curcuminoids and chalcones (**4**, **5**, **8**, **9**, **11**, **12**, and **13**) for 24 h, and cell viability was determined. Considering the IC50 values, compounds **5**, **9**, and **13** exhibited the highest values, indicating that the cells were more tolerant to these samples. IC50 values and their concentrations used in the experiments are listed in Table 2. Rasagiline, an MAO-B inhibitor, was used as a positive control for the experiments.

### 3.3. Cyclic C_5_-Curcuminoids and Chalcones Decrease the Production of ROS by Increasing Antioxidant Defense

6-OHDA treatment of differentiated cells significantly increased ROS levels, indicating the induction of oxidative stress, a key feature of PD. Compounds **5**, **9**, **12**, and **13** significantly decreased ROS production compared to 6-OHDA (135.43 ± 7.2%) and control (100 ± 8.06%). Moreover, compound **13** (55.16 ± 4.56%), which was the most potent curcuminoid, was more efficient in reducing ROS levels than rasagiline (82.25 ± 3.68%) (Figure 3A). However, the extreme ROS-reducing capacity of compound **13** may lead to a contrary biological phenomenon, known as reductive stress, a harmful process for the cells.

As ROS levels were altered in the presence of the sample compounds, we also assessed the TAC. Our findings revealed that the TAC exhibited changes that were inversely related to the alterations in ROS levels. Compounds **5**, **9**, **12**, and **13** significantly elevated TAC levels compared to 6-OHDA (15 ± 2.11 µM) and control (38.33 ± 3.54 µM). Compound **13** (147.5 ± 4.77 µM) was the most effective and was more efficient than rasagiline (92.36 ± 3.12 µM) (Figure 3B). However, compounds **4** and **11** also increased TAC relative to 6-OHDA treatment (Figure 3B). Appendix A presents the results obtained from two additional treatment concentrations of the test compounds.

### 3.4. Cyclic C_5_-Curcuminoid and Chalcone Derivatives Influence the Activity of the Antioxidant Enzymes GPx and SOD

The activity of antioxidant enzymes was also determined based on TAC results. The 6-OHDA treatment of differentiated SH-SY5Y cells significantly decreased GPx activity but not SOD activity (Figure 4A,B). Compounds **5**, **9**, **12**, and **13** demonstrated increased GPx activity, while compound **11** exhibited lower efficiency (Figure 4A). Rasagiline also effectively elevated GPx levels, with only compound **12** surpassing rasagiline in GPx levels (Figure 4A). Furthermore, compound **12** was the sole compound capable of enhancing GPx activity compared to the control.

In SOD activity measurements, we found that our sample compounds were less effective against SOD. Only compounds **5**, **12**, and **13** increased SOD activity with the same potency as that of rasagiline.

### 3.5. Cyclic C_5_-Curcuminoids 9, 12, 13 and Cyclic Chalcone 5 Inhibit 6-OHDA-Induced Iron Accumulation and Lipid Peroxidation

Iron accumulation in Parkinsonian neurons contributes to free radical production, thereby inducing oxidative stress and lipid peroxidation. Treatment with 6-OHDA significantly elevated the intracellular iron levels. Notably, all compounds, except compound **11**, demonstrated some capacity to reduce the iron content (Figure 5A). Compound **5** demonstrated the highest efficacy, as it was the sole compound capable of reducing iron levels (13.08 ± 0.74 µM iron/mg protein) below the control level (18.15 ± 0.59 µM iron/mg protein) (Figure 5A).

Thiol levels are a suitable marker of oxidative stress. The elevated thiol levels upon 6-OHDA treatment in differentiated SH-SY5Y cells indicated the initiation of lipid peroxidation (Figure 5B). Compounds **5**, **9**, **12**, and **13**, as well as rasagiline, reduced thiol levels compared with 6-OHDA. However, they did not differ significantly from that in the control group (Figure 5B). The findings indicate that the four derivatives may offer protection against lipid peroxidation. Notably, the other three compounds, **4**, **8**, and **11**, significantly reduced thiol levels (Figure 5B), suggesting that these compounds may interact with free thiol groups within the cells, resulting in their depletion.

MDA level, which is produced by lipid peroxidation, significantly increased upon 6-OHDA treatment of the differentiated SH-SY5Y cells. Compounds **5**, **9**, **12**, and **13** reversed the effect of 6-OHDA, albeit without reducing the MDA levels to those observed in the control group. Notably, compounds **4**, **8**, and **11**, which reduced intracellular total thiol levels, resulted in elevated MDA concentrations within the cells, indicating the deteriorating effects on the cells.

### 3.6. Cyclic C_5_-Curcuminoids and Cyclic Chalcones Affect Mitochondrial Function and Inhibit Apoptosis

Increased ROS production may lead to mitochondrial dysfunction, reduced energy production, the release of cytochrome c, and the activation of caspase-3, an effector caspase, leading to apoptosis. The 6-OHDA treatment of differentiated SH-SY5Y cells significantly reduced ATP levels, increased cytochrome c release, and enhanced apoptosis (Figure 6A,B). Compounds **5**, **9**, **12**, and **13** restored energy production to the control level (Figure 6A). In parallel with these findings, the aforementioned compounds significantly decreased cytochrome c concentration and caspase-3 activation (Figure 6A–C). Compound **5** was proven to act the most effectively in the in vitro 6-OHDA-induced neuronal damage model. The effect of rasagiline was similar to that of compounds **5** and **13** (Figure 6). The observed change in the case of compounds **4**, **8** and **11** may not reflect the negative effects of the compounds on mitochondrial function, but rather the influence of 6-OHDA, as there was no significant difference between the 6-OHDA treatment and the 6-OHDA combined with compounds **4**, **8**, and **11**.

### 3.7. Cyclic C_5_-Curcuminoids 9, 12, 13 and Cyclic Chalcone 5 Exhibit Minimal Influence on Anti-Inflammatory Cytokine Production of the OHDA-Induced Differentiated SH-SY5Y Cells

Inflammation is a crucial process in the progression of PD. The secretion of pro-inflammatory and/or anti-inflammatory cytokines influences the function of microglia, which serve as the brain’s immune cells, and can either exacerbate or mitigate neurodegeneration. 6-OHDA treatment of differentiated SH-SY5Y cells did not alter IL-4 levels but significantly decreased the release of IL-10 (Figure 7A,B). Four sample compounds, **5**, **9**, **12**, and **13**, caused minor elevations in both IL-4 and IL-10 levels, similar to the effects of rasagiline on anti-inflammatory cytokines (Figure 7A,B). Interestingly, compounds **8** and **11**, akin to 6-OHDA, significantly decreased IL-10 secretion from the neurons (Figure 7B).

Based on these findings, it cannot be definitively concluded that the examined chalcones and curcuminoids possess significant anti-inflammatory properties. Their effect is limited to a modest enhancement in the production and release of the anti-inflammatory cytokines IL-4 and IL-10.

### 3.8. Cyclic C_5_-Curcuminoids and Cyclic Chalcones Reduce Pro-Inflammatory Cytokine Release of the OHDA-Induced Differentiated SH-SY5Y Cells

The secretion levels of the pro-inflammatory cytokines IL-6, IL-8, and TNF-α were also assessed. Treatment of differentiated SH-SY5Y cells with 6-OHDA resulted in a significant increase in the release of all three pro-inflammatory cytokines (Figure 8). We also observed that the sample compounds had few specific effects on these cytokines. Compounds **4**, **5**, **8**, and **9** significantly decreased IL-6 secretion compared to 6-OHDA and the control; however, compounds **11**, **12**, and **13**, as well as rasagiline, increased its secretion compared to the control (Figure 8A).

In the case of IL-8, compounds **5**, **8**, **9**, and **13**, along with rasagiline, significantly reduced IL-8 secretion compared to 6-OHDA treatment (Figure 8B). Conversely, compound **4** notably increased IL-8 release from the neurons (Figure 8B). Regarding TNF-α, compounds **4**, **5**, **9**, **12**, and **13**, but not rasagiline, significantly decreased TNF-α secretion (Figure 8C).

According to these findings, only compounds **5** and **9** effectively reduced the levels of all three pro-inflammatory cytokines, while the effects of the other five compounds remain controversial.

### 3.9. Cyclic C_5_-Curcuminoids 9, 12, 13, and Cyclic Chalcone 5 Reduce the Release of Fractalkine from the OHDA-Induced Differentiated SH-SY5Y Cells

Soluble fractalkine, an inflammatory chemokine, contributes to the activation of microglia in PD. Analysis of fractalkine levels showed that compounds **5**, **9**, **12**, and **13** significantly reduced the release of soluble fractalkine from differentiated SH-SY5Y cells compared to 6-OHDA treatment (Figure 9). Conversely, compound 8 markedly increased fractalkine secretion, while compounds **4** and **11**, along with rasagiline, did not affect fractalkine production compared to 6-OHDA (Figure 9).

The results indicate that compounds 5 and 9 effectively reduced the levels of all four inflammatory proteins (IL-6, IL-8, TNF-α, and fractalkine), thereby suggesting their potential role as anti-inflammatory agents.

## 4. Discussion

Parkinson’s disease (PD) progression includes the overproduction of reactive oxygen species, iron accumulation, lipid peroxidation, and mitochondrial dysfunction, resulting in neuronal cell death [3]. The overactivation of microglia, which are the innate immune cells of the brain, contributes to the development and deterioration of the disease by releasing pro-inflammatory cytokines and promoting inflammation [44,45].

The utilization of plant-derived molecules and their synthetic derivatives is gaining ground in the treatment of difficult-to-treat diseases, mainly as complementary therapies [46]. The prevalence of neurodegenerative diseases (such as Parkinson’s disease, Alzheimer’s disease, Huntington’s disease, and amyotrophic lateral sclerosis) is steadily increasing worldwide, placing an ever-greater burden on society [47]. Most therapies aim only to relieve symptoms and slow the progression of the disease, mainly due to the complex background of neurodegeneration [48]. New approaches and strategies are being developed, notably stem cell treatment and gene therapy [49,50].

Turmeric (*Curcuma longa* L.) contains curcumin, the most abundant curcuminoid. The other three natural curcuminoids in Curcuma are demethoxycurcumin, bisdemethoxycurcumin, and cyclo-curcumin [18,51,52]. Curcumin has been shown to possess neuroprotective potential in PD models [53,54,55,56], and exhibits antibacterial, antifungal, and anticancer activities [57,58]. Recently, it was reported that the ethanol extract of Curcuma has free radical scavenging capacity [59], suggesting a beneficial effect of curcumin on reactive oxygen species. Since oxidative stress is one of the underlying causes of the progression of PD, these observations support that curcuminoids are potential antioxidant molecules and also provide neuroprotection [29,57,60]. However, no experimental data are available on the effects of synthetic cyclic C_5_-curcuminoids in a Parkinson’s model.

In this study, we investigated novel synthetic cyclic C_5_-curcuminoids for their potential antioxidant activity within an in vitro neurodegeneration model using ATRA-differentiated SH-SY5Y cells, where oxidative stress was induced by 6-OHDA treatment. Previous studies have demonstrated [37,61,62,63] that 6-OHDA treatment leads to increased iron accumulation. This occurs due to the elevated expression of iron importer transferrin receptor-1 and heme oxygenase-1, which facilitates the release of free iron into the cytosol through heme degradation. Additionally, 6-OHDA slightly decreases the expression of the ferroportin iron exporter, potentially contributing to iron accumulation [37]. Consequently, this enhances the reactive iron (Fe^2+^) content within the labile iron pool, potentially initiating the Fenton reaction and elevating ROS levels. Simultaneously, the auto-oxidation of 6-OHDA in cells, in the presence of oxygen, iron, or copper, leads to the generation of superoxide anion and hydrogen peroxide [64,65,66]. Furthermore, 6-OHDA inhibits Complex I and Complex IV in the mitochondrial respiratory chain, resulting in increased superoxide production [64,67,68].

Rasagiline, a MAO-B inhibitor known for its antioxidant and anti-apoptotic properties, was utilized as a positive control, providing neuroprotection [69,70]. Our investigation focused on the cellular antioxidant defense mechanisms, which involved assessing ROS production, evaluating TAC, and measuring the activity of GPx and SOD. These findings provide insights into the level of antioxidant protection in cells, attributed to small molecules and antioxidant enzymes [71].

6-OHDA treatment of differentiated SH-SY5Y cells induced ROS production but decreased TAC and GPx activity [65,72,73]. The addition of cyclic sample compounds to 6-OHDA-treated cells separated the participating derivatives into two distinct groups. Compounds **5**, **9**, **12**, and **13** significantly reduced ROS levels while enhancing TAC, GPx, and SOD activity. Notably, compounds **9** and **12** demonstrated higher efficacy compared to rasagiline, with the exception of SOD activity. Additionally, in the case of compound **12**, both SOD and GPx activities were significantly increased. The observed extreme ROS-reducing capacity of compound **13** may lead to a contrary biological phenomenon, known as reductive stress, which can result in an excess of reducing agents, mitochondrial dysfunction, metabolic disruption, and potentially increased inflammation [74,75,76]. The SOD enzymes, MnSOD and Zn/CuSOD, are essential for eliminating superoxide anions, which subsequently produce hydrogen peroxide. This hydrogen peroxide is then neutralized by GPx enzymes. Together with the thioredoxin system, these enzymes form the primary defense against oxidative damage [77]. In contrast, compounds **4**, **8**, and **11** in the second group did not alter ROS levels or GPx and SOD activity. However, compounds **4** and **11** did show a modest increase in TAC levels. This effect may be attributed to the direct free-radical scavenging properties of curcuminoids [59]. The ability of curcumin and curcuminoids to reduce ROS was also noted in SH-SY5Y cells infected with SARS-CoV-2 [78].

Hydroxyl radicals are produced by free iron in the cytoplasm through the Fenton reaction [77]. When excess free radicals reach macromolecules, they can inflict significant damage, leading to lipid and protein peroxidation and posing a risk to DNA integrity [79,80]. In differentiated SH-SY5Y cells, 6-OHDA increased iron content [81]. Notably, all compounds reduced iron levels, though at different rates. Compounds **5**, **9**, **12**, and **13** lowered the iron content to the control level, while cells treated with compounds **4** and **8** exhibited significantly higher iron levels than the control, indicating their reduced effectiveness. Synthetic cyclic C_5_-curcuminoids effectively diminished iron accumulation, thereby contributing to a reduction in ROS production [9].

Based on the intracellular iron content, the investigated test molecules can be separated into two groups. One plausible chemical explanation for the differentiation between the two groups of tested compounds is the ability of Fe(III) to form complexes with phenolic ligands, as seen in derivatives **5**, **9**, **12**, and **13** of the first group. A recent review article highlighted the interaction of Fe(III) with natural antioxidants, such as the Fe(III)–curcumin complex [82]. The authors concluded that curcumin has the capacity to sequester free iron, thereby preventing its detrimental accumulation within living cells. It is highly probable that, in our experiments, cells could be shielded from the accumulation of free iron through this chemical binding, resulting in neuroprotection in the case of phenolic compounds **5**, **9**, **12**, and **13**. Conversely, the complex-forming reactivity is significantly diminished in the second group (nonphenolic compounds **4**, **8**, and **11**) due to methyl substitution blocking the phenolic ligand, thereby permitting the progression of neurodegeneration.

The thiol-containing molecules, like glutathione (GSH), protein thiols, thioredoxin, α-lipoic acid, and *N*-acetylcysteine, are essential for eliminating free radicals, which can result in lipid peroxidation and subsequent cellular dysfunction [83]. Thiol measurements indicated that treatment with 6-OHDA resulted in elevated thiol levels, whereas compounds **5**, **9**, **12**, and **13** mitigated this effect. According to literature, an increase in thiol levels is indicative of the early stages of oxidative stress and lipid peroxidation [72]. During this phase, the sulfhydryl (-SH) groups of proteins, specifically cysteine and methionine amino acid residues, interact with the reactive species. This initial oxidative stress can lead to an increase in intracellular thiol levels, which subsequently decline as the concentration of ROS rises and thiol consumption, particularly of GSH, intensifies to protect the cells [84,85]. In our study, the measurement of thiols encompasses not only protein thiols and GSH but also other thiol-containing molecules such as thioredoxin. Alpha-lipoic acid, another thiol-containing compound, functions as a neuroprotective agent in PD [61] and can enhance GSH synthesis. Similarly, *N*-acetylcysteine is a thiol-containing antioxidant that reduces oxidative stress and promotes GSH synthesis primarily through deacetylation, which supplies cysteine for GSH synthesis [86]. It also exerts beneficial effects on dopaminergic neurons in PD [87]. Based on the total thiol measurements, we posit that the in vitro neural damage model system represents the early stage of oxidative stress, which is mitigated by compounds **5**, **9**, **12**, and **13** but not by compounds **4**, **8**, and **11**. In the latter case, we suggest that these cyclic C_5_-curcuminoids exhibit thiol-binding activity rather than ROS scavenging activity [88,89,90].

Analysis of the TAC and thiol concentrations suggests that the observed effects of the tested compounds may appear contradictory. Total antioxidant capacity measurements determine the capacity of small molecule antioxidants and antioxidant enzymes, such as SOD, GPx, and catalase. 6-OHDA treatment of differentiated SH-SY5Y cells decreased the TAC. Although the thiol level was elevated, GPx activity significantly decreased. In the case of compounds **5**, **9**, **12**, and **13**, the thiol content was similar to that of the control, but TAC was significantly increased. This can be explained by the increased GPx activity in the case of compounds **9** and **12**, and the elevated SOD activity in the case of compounds **5** and **13**. Moreover, in the case of compound **12**, in which the highest TAC level was measured, both SOD and GPx activities were significantly increased. We cannot exclude the direct free radical scavenging capability of the test compounds, which can also increase TAC [89,90,91]. The proposed effects of compounds **5**, **9**, **12**, and **13** are described in Figure 4.

We evaluated the levels of MDA, a marker for lipid peroxidation. The data reveal that 6-OHDA treatment triggered lipid peroxidation, whereas compounds 5, 9, 12, and 13 alleviated the impact of 6-OHDA, although they did not bring MDA levels down to those of the control group. Remarkably, compounds **4**, **8**, and **11**, which led to a reduction in intracellular total thiol levels, resulted in an increase in MDA concentrations within the cells. However, these increases were not statistically significant compared to 6-OHDA alone. Nonetheless, these observations suggest that compounds **4**, **8**, and **11** merit further investigation as potential anticancer agents.

Increased production of ROS also impacts mitochondrial function. Free radicals target the mitochondrial membrane, leading to a reduction in ATP production and the subsequent release of cytochrome c, which activates apoptosis [7,92]. In our neurodegeneration model, ATP levels were significantly reduced following 6-OHDA treatment, indicating a decrease in oxidative phosphorylation [93]. Compounds **5**, **9**, **12**, and **13** were able to maintain ATP production at control levels, whereas treatment of 6-OHDA-induced SH-SY5Y cells with compounds **4**, **8**, and **11** did not mitigate the adverse effects of 6-OHDA on ATP levels.

The release of cytochrome c is contingent upon the integrity of the mitochondrial membrane. ROS can target membrane proteins, resulting in the enhanced release of cytochrome c or the opening of the permeability transition pore complex (PTPC), which increases the permeability of the outer membrane and prompts cytochrome c release [94,95]. It has been demonstrated that curcuminoids can induce apoptosis by promoting cytochrome c release and activating caspase-3 in cancer cells [96,97,98]. In our in vitro neurodegeneration model, 6-OHDA was found to elevate cytochrome c levels and increase relative caspase-3 activity in neurons [99]. Compounds **5**, **9**, **12**, and **13**, similar to rasagiline, effectively reduced cytochrome c concentrations and significantly decreased caspase-3 activity compared to 6-OHDA, highlighting the anti-apoptotic properties of these cyclic C_5_-curcuminoids. Conversely, compounds **4**, **8**, and **11** did not affect cytochrome c levels or caspase-3 activity relative to 6-OHDA, indicating a lack of anti-apoptotic effect on differentiated SHSY-5Y cells. Furthermore, compounds **5** and **9** emerged as the most potent in reducing the concentration and activity of apoptosis-related proteins.

Based on the overall results, the test compounds utilized in this study, specifically cyclic C_5_-curcuminoids and chalcones, can be biologically categorized into two distinct groups. The first group of compounds (**5**, **9**, **12**, and **13**) demonstrated efficacy in mitigating oxidative stress, reducing oxidative damage, and enhancing mitochondrial function. Additionally, this group was effective in decreasing apoptosis. Conversely, the second group (compounds **4**, **8**, and **11**) did not exhibit success in inhibiting oxidative stress or apoptosis. The biological differentiation between the two groups corresponded with their chemical classification. The first group comprised phenolic compounds (**5**, **9**, **12**, and **13**), whereas the second group consisted of nonphenolic derivatives (**4**, **8**, and **11**). The following chemical explanations for the separation of the tested compounds can be proposed:

In our previous study [100], we demonstrated that similar compounds are capable of participating in catalytic hydrogen transfer reactions. Such hydrogen transfer reactions, involving enzymes and other peptides, are ubiquitous in biological systems [101,102]. The hydrogen-donating capabilities of our model compounds **5**, **9**, **12**, and **13** in biochemical hydrogen-transfer reactions are attributed to their potential structural rearrangements, as illustrated in Figure 5.

Compounds **5**, **9**, **12**, and **13**, while giving off hydrogen, were oxidized to their corresponding quinoidal products **14**–**20**. Compound **5**, for example, was ortho quinone **15,** compound **9** was para quinone **18**, **12** to **19**, and derivatives **13** to **20**. The pronounced hydrogen donor property is responsible for several of the aforementioned beneficial effects, indicating that the first group of compounds (**5**, **9**, **12**, and **13**) may serve as potential neuroprotective agents. The group of hydrogen-donor phenolic compounds (**5**, **9**, **12**, and **13**) and their quinoidal products formed in hydrogen-transfer reactions (**14**–**20**) mimic the natural quinone–hydroquinone equilibrium. One of our hypotheses posits that the potential reduction of oxidized derivatives **14**–**20** back to compounds **5**, **9**, **12**, and **13**, utilizing physiological cellular hydrogen sources, may result in enhanced cellular protection until the tested compounds are metabolized. This hypothesis, however, requires further investigation and validation.

In contrast, the methyl substitution of phenolic OH groups with methoxy groups in compounds **4**, **8**, and **11** in the second group prevents such rearrangements, eliminating any potential for hydrogen donor activity. Moreover, these methoxy substituents promote thiol addition, known as the thia Michael reaction, over the benzylidene double bonds. These derivatives (**4**, **8**, and **11**) are associated with cellular toxicity rather than protection, which contributes to their harmful impact on cell viability.

Inflammation in PD is driven by neurons, microglia, brain immune cells, and astrocytes, all of which contribute to the inflammatory process [103,104,105]. Pro-inflammatory cytokines such as IL-6, TNF-α, and IL-8, along with chemokines like fractalkine and other biomolecules like glutamate, can activate microglia released by dopaminergic neurons in PD [106,107]. As PD progresses, microglial overactivation intensifies, worsening the disease [44,108]. Conversely, anti-inflammatory cytokines like IL-4 can encourage the anti-inflammatory M2 type of microglia [108]. Thus, IL-4 released by dopaminergic neurons in PD may offer neuroprotection [109]. Additionally, IL-10 modulates microglial activity, enhances phagocytic activity, and provides neuroprotection [110].

Our findings indicated that treating differentiated SH-SY5Y cells with 6-OHDA did not affect IL-4 levels but did increase the release of the anti-inflammatory cytokine IL-10. Compounds **5**, **9**, **12**, and **13**, akin to rasagiline, minimally enhanced the secretion of IL-4, a pattern also observed with IL-10; therefore, it cannot be definitively concluded that the examined chalcones and curcuminoids possess significant anti-inflammatory properties. Compounds **4**, **8**, and **11** did not alter IL-4 levels and, similar to 6-OHDA, reduced IL-10 release from neurons.

An analysis of the concentrations of the pro-inflammatory cytokines IL-6, IL-8, and TNF-α did not yield consistent results when compared with anti-inflammatory cytokines. Notably, compounds **4**, **5**, **8**, and **9** significantly reduced IL-6 secretion, whereas compounds **11**, **12**, and **13**, along with rasagiline, increased its levels. IL-8 levels were inversely correlated with those of anti-inflammatory cytokines. Compounds **5**, **9**, and **13** significantly reduced IL-8 release, whereas compounds **4**, **8**, and **11** elevated its level. TNF-α, a key mediator of inflammation, contributes to neurotoxicity in PD [111]. Its levels present a distinct perspective on the effects of the examined curcuminoids. Compounds **4**, **5**, **9**, **12**, and **13** decreased the TNF-α levels secreted by neurons, whereas compounds **8** and **11** and, interestingly, rasagiline did not.

Fractalkine, an uncommon chemokine, has been discovered in the central nervous system [112]. Membrane-bound fractalkine in the neurons maintains the resting state of microglia via the CX3CR1 fractalkine receptor expressed by the microglia [113]. On the other hand, the soluble fractalkine released from the membrane-bound form triggers microglial activation. It has also been reported that fractalkine levels increase in PD [114]. 6-OHDA treatment significantly increased fractalkine levels, whereas compounds **5**, **9**, **12**, and **13**, but not rasagiline, reduced fractalkine release from the differentiated SH-SY5Y cells. The findings indicate that cyclic C_5_-curcuminoids and chalcones confer neuroprotection by diminishing the secretion of soluble fractalkine, thereby contributing to the maintenance of microglial quiescence and attenuating the release of pro-inflammatory cytokines from microglia.

Interestingly, the effects of the sample compounds on inflammation could not be distinctly categorized into the previously mentioned two groups. While the anti-inflammatory properties of the first group of tested compounds are characterized by slightly elevated levels of anti-inflammatory cytokines and increased soluble fractalkine secretion, the pro-inflammatory cytokine measurements exclude compounds **12** and **13** due to their inflammation-promoting effects.

## 5. Conclusions

Based on the findings of this study, it can be concluded that four of the seven derivatives investigated, specifically cyclic chalcone 5 and synthetic C_5_-curcuminoids **9**, **12**, and **13**, demonstrated antioxidant and anti-apoptotic effects in our in vitro model of neurodegeneration. Nevertheless, the evaluation of anti-inflammatory effects led to the exclusion of compounds **12** and **13**, as compound **12** was associated with an increase in IL-6 and IL-8 pro-inflammatory cytokines, and compound **13** was linked to elevated IL-6 secretion. In addition, compound **13** markedly diminished the levels of ROS, which can induce reductive stress, a harmful condition for cells. Although these test compounds show promise as drug candidates, further in vitro investigation of cyclic chalcone **5** and synthetic C_5_-curcuminoid **9** is necessary to assess their impact on microglia, which play a critical role in disease progression. Subsequently, formulation and in vivo studies using an animal model of PD are required to validate their therapeutic efficacy.

## Data Availability

The original contributions presented in this study are included in the article/Appendix A. Further inquiries can be directed to the corresponding author.

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
