# Peer review of "Synthetic Cyclic C5-Curcuminoids Increase Antioxidant Defense and Reduce Inflammation in 6-OHDA-Induced Retinoic Acid-Differentiated SH-SY5Y Cells"

_antioxidants, 2025, doi:10.3390/antiox14091057_

Round 1

Reviewer 1 Report

In the manuscript entitled “Synthetic cyclic C5-curcuminoids increase antioxidant defense and reduce inflammation in 6-OHDA-induced retinoic acid-differentiated SH-SY5Y cells” the authors evaluated the effects of two chalcones and five synthetic C5 cyclic curcuminoids in an in vitro model of neurodegeneration (SH-SY5Y cells treated with 6-OHDA). The results showed that one cyclic chalcone (compound 5) and three curcuminoids (compounds 9, 12, 13) reduced oxidative stress and apoptosis. Specifically, compounds 5 and 9 also decreased the production of pro-inflammatory cytokines and increased anti-inflammatory cytokines. These compounds therefore show therapeutic potential as antioxidant, anti-apoptotic, and anti-inflammatory agents for the treatment of Parkinson's disease.

The article addresses a topic of great current interest in neuroscience: the use of synthetically modified natural compounds to counteract the pathogenic mechanisms of Parkinson's disease, particularly oxidative stress and inflammation. The authors present a well-structured in vitro study using a well-established cellular model (differentiated SH-SY5Y cells treated with 6-OHDA) to simulate neurodegeneration. The experimental approach is robust: multiple biochemical parameters were assessed, including ROS production, antioxidant enzyme activity, apoptosis, and cytokine release. The results highlight the antioxidant, anti-inflammatory, and anti-apoptotic potential of two compounds in particular (5 and 9), which appear to be promising lead molecules for the development of new treatments. However, to strengthen the scope and applicability of the results, some aspects could be improved or expanded.

Minor points.

  1. I recommend that the authors extend the experimental model. Indeed, the study relies exclusively on an in vitro model. Although valid for initial screening, the lack of in vivo data limits the applicability of the findings. I suggest that the authors consider future experiments on animal models of Parkinson's disease, such as MPTP or 6-OHDA in rats or mice, to evaluate the neuroprotective effects at the behavioral, molecular, and histological levels.
  2. Since one of the main challenges in the use of curcuminoids is their poor bioavailability and limited ability to cross the blood-brain barrier (BBB), it would be useful to include predictive in vitro studies (e.g., BBB models with endothelial cells) or permeability assays (e.g., PAMPA-BBB). Furthermore, I suggest the authors test synthetic compounds for their metabolic stability and ADME profile.
  3. To elucidate the mechanisms of action of the most effective compounds, I recommend the authors to analyze key signaling pathways involved in the response to oxidative stress and apoptosis (e.g., Nrf2, NF-κB, MAPK).
  4. Overall, the references are appropriate and well distributed between the introduction and the discussion. However, the use of some dated sources is noteworthy (e.g., references [2], [3], [5], [6] prior to 2010), particularly in the paragraphs dealing with the pathophysiology of Parkinson's disease. Considering the recent expansion of the literature on these topics, it would be appropriate to update some citations with more recent studies (from 2020 onwards).
  5. I recommend revising the manuscript to eliminate repetitions (e.g., “suggest that,” “indicate that”) and strengthen the scientific style, perhaps with the support of professional language editing.

In conclusion, the work is well structured and presents interesting and potentially innovative results. However, to achieve publication in a high-impact journal, it is desirable to:

  • Integrate the work with more translational experiments (in vivo, pharmacokinetics);
  • Update the bibliographic sources;
  • Strengthen the academic language of the text.

With these revisions, the article has the potential to significantly contribute to the development of new therapeutic approaches in the treatment of Parkinson's disease.

Author Response

Reviewer 1.

Major comments

In the manuscript entitled “Synthetic cyclic C5-curcuminoids increase antioxidant defense and reduce inflammation in 6-OHDA-induced retinoic acid-differentiated SH-SY5Y cells” the authors evaluated the effects of two chalcones and five synthetic C5 cyclic curcuminoids in an in vitro model of neurodegeneration (SH-SY5Y cells treated with 6-OHDA). The results showed that one cyclic chalcone (compound 5) and three curcuminoids (compounds 9, 12, 13) reduced oxidative stress and apoptosis. Specifically, compounds 5 and 9 also decreased the production of pro-inflammatory cytokines and increased anti-inflammatory cytokines. These compounds therefore show therapeutic potential as antioxidant, anti-apoptotic, and anti-inflammatory agents for the treatment of Parkinson's disease.

The article addresses a topic of great current interest in neuroscience: the use of synthetically modified natural compounds to counteract the pathogenic mechanisms of Parkinson's disease, particularly oxidative stress and inflammation. The authors present a well-structured in vitro study using a well-established cellular model (differentiated SH-SY5Y cells treated with 6-OHDA) to simulate neurodegeneration. The experimental approach is robust: multiple biochemical parameters were assessed, including ROS production, antioxidant enzyme activity, apoptosis, and cytokine release. The results highlight the antioxidant, anti-inflammatory, and anti-apoptotic potential of two compounds in particular (5 and 9), which appear to be promising lead molecules for the development of new treatments. However, to strengthen the scope and applicability of the results, some aspects could be improved or expanded.

Minor points

  1. I recommend that the authors extend the experimental model. Indeed, the study relies exclusively on an in vitro model. Although valid for initial screening, the lack of in vivo data limits the applicability of the findings. I suggest that the authors consider future experiments on animal models of Parkinson's disease, such as MPTP or 6-OHDA in rats or mice, to evaluate the neuroprotective effects at the behavioral, molecular, and histological levels.

Thank you for your comment. Conducting in vivo experiments using Parkinson's models is essential to validate the antioxidant, anti-apoptotic, and anti-inflammatory effects of the molecules being studied. Prior to performing these experiments, we aimed to determine whether these compounds influenced microglial function, as inflammation caused by microglia significantly contributes to the progression of Parkinson's disease. We are currently examining this interaction within neuron-microglia co-cultures. If successful, the deleterious effects of microglia on PD could be mitigated. Following this, the selected curcuminoids must be appropriately formulated to enhance their permeability across the blood-brain barrier, thereby facilitating their application in vivo.

Since one of the main challenges in the use of curcuminoids is their poor bioavailability and limited ability to cross the blood-brain barrier (BBB), it would be useful to include predictive in vitro studies (e.g., BBB models with endothelial cells) or permeability assays (e.g., PAMPA-BBB). Furthermore, I suggest the authors test synthetic compounds for their metabolic stability and ADME profile.

Curcuminoids can traverse the plasma membrane owing to their lipophilic nature (10.3389/fnut.2022.1040259). However, their bioavailability and ability to penetrate the blood-brain barrier remain limited. Interestingly, cyclic C5-curcuminoids exhibit improved permeability (10.1016/j.ejps.2022.106184). We did not perform BBB permeability assays using the test compounds. However, three of our synthetic compounds successfully traversed the BBB, one of which is a structural analog closely related to compound 13 discussed in this manuscript, differing by the presence of a trifluoromethyl group instead of a hydroxyl group and a hydrogen atom in place of a methoxy group (10.1016/j.ejps.2022.106184). Additional structural modification, conjugation, or the use of nanotechnological formulations with various nanoparticles can enhance BBB penetration (10.3390/ijms22010196; 10.3389/frdem.2023.1222708; 10.1016/j.ijpharm.2017.05.015; 10.3390/jcm9020430; 10.1007/978-981-99-7731-4_6).

In addition to exhibiting significantly enhanced anticancer activity, cyclic C5-curcuminoids demonstrate superior stability compared to curcumin, as evidenced by their reduced susceptibility to hydrolysis and metabolism (10.1016/j.molstruc.2019.127661; 10.1016/j.ejmech.2018.02.008; 10.1007/s00044-017-2056-x; 10.3390/molecules29225321; 10.1021/acsomega.8b02625; 10.53555/ecb/2023.12.si5a.0159). Furthermore, these compounds are characterized by excellent tolerability in mammals, as reported by Yuan et al. (10.1016/j.ejmech.2014.03.012) and Anchoori et al. (10.1016/j.ccr.2013.11.001).

We are currently working on identifying an optimal structure that exhibits high activity and demonstrates significant permeability and stability.

  1. To elucidate the mechanisms of action of the most effective compounds, I recommend the authors to analyze key signaling pathways involved in the response to oxidative stress and apoptosis (e.g., Nrf2, NF-κB, MAPK).

Thank you for your comment. This study aimed to determine whether the test compounds can alter cytokine secretion in differentiated SH-SY5Y cells. The subsequent aim is to evaluate whether these compounds affect microglial function, given that microglia-induced inflammation significantly contributes to the progression of Parkinson's disease. We are currently investigating this interaction in neuron-microglia co-cultures. In this co-culture system, we examine the NF-κB pathway in both neurons and microglia. Additionally, the Nrf2-Keap-1 system is being studied in co-cultured neurons, as microglia can contribute to oxidative stress. We believe that the results of signal transduction from co-cultured cells offer deeper insights into the mechanisms of action of curcuminoids.

  1. Overall, the references are appropriate and well distributed between the introduction and the discussion. However, the use of some dated sources is noteworthy (e.g., references [2], [3], [5], [6] prior to 2010), particularly in the paragraphs dealing with the pathophysiology of Parkinson's disease. Considering the recent expansion of the literature on these topics, it would be appropriate to update some citations with more recent studies (from 2020 onwards).

Thank you for the suggestion. The references have been updated accordingly.

  1. I recommend revising the manuscript to eliminate repetitions (e.g., “suggest that,” “indicate that”) and strengthening the scientific style, perhaps with the support of professional language editing.

Thank you for the comment. The entire manuscript has been edited; please refer to the highlighted sections.

All changes have been highlighted in the manuscript.

Reviewer 2 Report

The manuscript should be improved by editing carefully. 

The authors investigated the effects of new curcumin derivatives on oxidative stress and inflammation in 6-OHDA-induced retinoic acid differentiated SH-SY5Y cells. They detected some perimeters, such as ROS, total antioxidant capacity, antioxidant enzyme activity, thiol and ATP levels, caspase-3 activity, and cytokine release, after treatment with the test compounds. They found that one cyclic chalcone (compound 5) and three synthetic cyclic C5-curcuminoids (compounds 9, 12, and 13) decreased oxidative stress and apoptosis in the in vitro model of neurodegeneration. Compounds 5 and 9 were also effective in reducing the secretion of pro-inflammatory cytokines (IL-6, IL-8, and TNF-α) and enhancing the release of anti-inflammatory cytokines (IL-4 and IL-10). These results suggest that these two compounds exhibit potential antioxidant, anti-apoptotic, and anti-inflammatory properties, rendering them promising candidates for drug development. However, some concerns are raised that reduce the quality of this manuscript.

  1. The Introduction section of the manuscript had 12 short paragraphs, which made it appear quite disjointed. It needs to be reorganized.
  2. The Materials and Methods section needs to be rewritten to ensure it is original and specific to this manuscript.
  3. Results: (1) Figure 3: At least two doses of each compound are needed to determine their effectiveness, for example, one sub-threshold dose, one effective dose. (2) Completing only the cell-based assay is not sufficient to conclude that these two compounds will be effective in animals, let alone in humans in the future. (3) After obtaining the results shown in Figure 3, it would be more appropriate to use the effective compounds at different doses (e.g., one sub-threshold and one effective dose) in subsequent experiments to further evaluate their efficacy. However, the current study continued to test all compounds, even those that appeared ineffective in the initial experiments.
  4. The Discussion section contains 19 paragraphs, which made it appear too fragmented. It needs to be rewritten and reorganized for better coherence.

Author Response

Reviewer 2.

Major comments

The manuscript should be improved by editing carefully. 

Thank you for the advice. The entire manuscript has been reorganized and rewritten accordingly.

Detailed comments

The authors investigated the effects of new curcumin derivatives on oxidative stress and inflammation in 6-OHDA-induced retinoic acid differentiated SH-SY5Y cells. They detected some perimeters, such as ROS, total antioxidant capacity, antioxidant enzyme activity, thiol and ATP levels, caspase-3 activity, and cytokine release, after treatment with the test compounds. They found that one cyclic chalcone (compound 5) and three synthetic cyclic C5-curcuminoids (compounds 9, 12, and 13) decreased oxidative stress and apoptosis in the in vitro model of neurodegeneration. Compounds 5 and 9 were also effective in reducing the secretion of pro-inflammatory cytokines (IL-6, IL-8, and TNF-α) and enhancing the release of anti-inflammatory cytokines (IL-4 and IL-10). These results suggest that these two compounds exhibit potential antioxidant, anti-apoptotic, and anti-inflammatory properties, rendering them promising candidates for drug development. However, some concerns are raised that reduce the quality of this manuscript.

  1. The Introduction section of the manuscript had 12 short paragraphs, which made it appear quite disjointed. It needs to be reorganized.

Thank you for your comment. The introduction has been rewritten, reorganized, and shortened to improve clarity. We hope that this will make it more reader-friendly and informative.

  1. The Materials and Methods section needs to be rewritten to ensure it is original and specific to this manuscript.

Thank you for the comment. The Materials and Methods section has been modified accordingly. Methodological references to our previous studies have been added to this section of the manuscript.

  1. Results: (1) Figure 3: At least two doses of each compound are needed to determine their effectiveness, for example, one sub-threshold dose, one effective dose. (2) Completing only the cell-based assay is not sufficient to conclude that these two compounds will be effective in animals, let alone in humans in the future. (3) After obtaining the results shown in Figure 3, it would be more appropriate to use the effective compounds at different doses (e.g., one sub-threshold and one effective dose) in subsequent experiments to further evaluate their efficacy. However, the current study continued to test all compounds, even those that appeared ineffective in the initial experiments.

Thank you for your comment. At the outset of the experiments involving ROS and TAC measurements, we evaluated three different concentrations of the test compounds. Based on this preliminary assessment, we selected one effective concentration for subsequent experiments. However, although higher concentrations appeared to be more effective based on these results, the standard deviation increased significantly. This suggests that determining the efficiency of the compounds near the IC50 values is challenging. The results of ROS and TAC measurements obtained with the initial three concentrations are presented in Supplementary Figure 3. The initial concentrations of the test compounds are listed in Table 1 in the Materials and Methods section. The concentrations used for further investigation are presented in Table 2 in the Results section of this paper.

We further analyzed the compounds that demonstrated a lack of antioxidant activity to determine whether they possessed anti-apoptotic or anti-inflammatory properties independent of their antioxidant activity. The results indicate that Although compounds 4, 8, and 11 were not effective in the PD model, it may be beneficial to investigate their potential anti-cancer effects.

We will conduct further experiments on microglia co-cultured with neurons to assess the effectiveness of the test compounds in reducing the release of inflammatory mediators by microglia. These findings may alter the effective concentration. Subsequently, we will proceed with in vivo testing of the compounds, contingent on obtaining the appropriate formulation.

  1. The Discussion section contains 19 paragraphs, which made it appear too fragmented. It needs to be rewritten and reorganized for better coherence.

Thank you for the advice. The discussion section has been rearranged and rewritten for improved coherence. Please refer to the highlighted sections in the manuscript.

All changes have been highlighted in the manuscript.

Round 2

Reviewer 2 Report

The authors have adequately addressed the questions and revised the manuscript. No additional comments.

The authors have adequately addressed the questions and revised the manuscript. No additional comments.

Author Response

Thank you for the approval.